# AutoSFX: Automatic Sound Effect Generation for Videos

Submission Id: 2603*

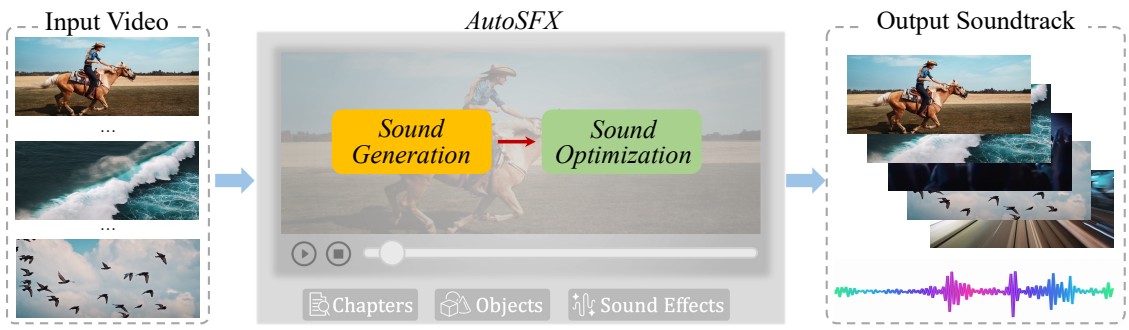

**Figure 1: Video in, sound effect out: our *AutoSFX* automatically generates sounds guided by visual information and further optimizes them to output a soundtrack with seamess transition and harmonious mixing. Our system consists of two modules, *i.e. Sound Generation* (§ 4) and *Sound Optimization* (§ 5). We also develop an interface (§ 6) for users to simplify sound design for video making and facilitate personalized requirements.**

## ABSTRACT

Sound Effect (SFX) generation, primarily aims to automatically produce sound waves for sounding visual objects in images or videos. Rather than learning an automatic solution to this task, we aim to propose a much broader system, *AutoSFX*, significantly applicable and less time-consuming, *i.e.* automating sound design for videos.Our key insight is that ensuring consistency between auditory and visual information, performing seamless transitions between sound clips, and harmoniously mixing sounds playing simultaneously, is crucial for creating a unified audiovisual experience. *AutoSFX* capitalizes on this concept by aggregating multi-modal representations by cross-attention and leverages a diffusion model to generate sound with visual information embedded. *AutoSFX* also optimizes the generated sounds to render the entire soundtrack for the input video, leading to a more immersive and engaging multimedia experience. We have developed a user-friendly interface for *AutoSFX* enabling users to interactively engage in the SFX generation for their videos with particular needs. To validate the capability of our vision-to-sound generation, we conducted comprehensive experiments and analyses using the widely recognized VEGAS and VGGSound test sets, yielding promising results. We also conducted a user study to evaluate the performance of the optimized soundtrack and the usability of the interface. Overall, the results revealed that our *AutoSFX* provides a viable sound landscape solution for making attractive videos.

## CCS CONCEPTS

• **Applied computing** → **Sound and music computing**; • **Human-centered computing** → **Interaction design**.

## KEYWORDS

Sound effect, sound design, sound generation, audiovisual consistency.

**ACM Reference Format:**
Anonymous Author(s). 2018. AutoSFX: Automatic Sound Effect Generation for Videos. In *Proceedings of Make sure to enter the correct conference title from your rights confirmation emai (Conference acronym 'XX)*. ACM, New York, NY, USA, 9 pages. https://doi.org/XXXXXXX.XXXXXXX

## 1 INTRODUCTION

Throughout the history of video making, sound effects (SFX) have played a crucial role in enhancing storytelling and creating immersive experiences for audiences [37]. The production of SFX involves identifying the essential visual objects and expressing the corresponding semantics with sounds. This process requires a deep understanding of audiovisual consistency as through exposure to a tremendous amount of visual-audio combinations, humans subconsciously learn the intricate correlations between visual and audio stimuli [14].

In recent years, significant advancements have been made in the automated SFX generation, realizing sounds for a single object in images or videos, such as dog barking, baby crying, and the explosion sounds of fireworks. Thanks to deep learning techniques and large-scale datasets (*e.g.*, AudioSet [15]), many models applied RNN, GAN, VQ-VAE, diffusion model, etc., to address the SFX generation task. However, relying solely on these strategies proves unsatisfactory for practical scenarios, *e.g.*, sound design for videos and VR games. We then pose the question – can computers further imitate the sound effect generation process like a sound designer, converting a video from concrete visual information to auditory

signals, *i.e.* elevating the SFX generation task to a comprehensive and practical sound design (SD) task? For example, a child playing in a garden should be synchronized with laughter, footsteps, bird chirping, etc., with all these sounds blending harmoniously and transiting seamlessly.

In this paper, we propose to compile the rich spatially and temporally audiovisual correlations into the prevalent sound generation regime. Similar to how humans construct a perceptual space by extracting information from visual objects, grouping objects by distance, and incorporating mixed sounds, our key insight is that visual information can further serve as strong learning signals for the SD task. For instance, by leveraging segmented visual objects, we can identify and localize the sound events; according to the different distances of visual objects, we can layer the mixed sounds.

Building on this insight, we introduce *AutoSFX*, a new SD pipeline that aims to achieve improved performance of sound generation and automated practical sound design for videos (*cf.* Fig. 1). Specifically, we first extract compact features with pixel-wise audiovisual features based on the Segment Anything Model (SAM) [24], which has recently proven to be highly effective in image segmentation tasks. We then employ adapters and cross-attention mechanisms to learn the correlation between audio and visual information, *i.e.* fusing pixel-wise audiovisual representations. For SFX generation, we leverage a spectrogram autoencoder to predict self-supervised auditory representation and a diffusion model to map visual information to latent representations. To further refine *AutoSFX*'s applicability to sound design tasks, we devise two modules: i) *Transition Module*, facilitates seamless transitions of generated sounds in alignment with changes in visual objects or scenes; ii) *Hierarchy Module*, aims to blend the sounds that playback simultaneously according to their corresponding depths. These modules are designed to circumvent the labor-intensive processes typically associated with text-prompted sound effect generation applications, such as Pika [1]. With *AutoSFX*, we are not only able to automate sound generation but also contribute to the field of sound design for videos.

To evaluate the effectiveness of our approach, we first undertook experiments on VEGAS [53] and VGGSound [4] to evaluate *AutoSFX*'s sound generation capabilities. We also conducted user studies to qualitatively evaluate the generated sound effects and the implemented interface. Experimental results demonstrate that our *AutoSFX* achieves a new state-of-the-art performance of sound generation and could provide promising results for videos featuring diverse objects and scenes. Our main contributions are:

- We propose *AutoSFX*, a deep learning-based system to tackle the problem of automatically generating sound effects for videos with seamless transition and harmonious mixing.
- We model the SD task based on the Segment Anything Model (SAM), leveraging adapter and cross attention in the early stage to perform audiovisual fusion. Specifically, we utilize a diffusion model to project visual information into the auditory space.
- Extensive experiments have validated the effectiveness of our *AutoSFX*, which exhibit solid performance gains on sound generation models, such as VEGAS and VGGSound,

and received positive feedback from both video creators and sound designers.

## 2 RELATED WORK

In this section, we first give a concise overview of the sound design industry. Then, we give a brief discussion on vision-to-audio techniques and the segment anything model (SAM) that pertains closely to our work.

### 2.1 Sound Design Industry

The use of sound in theater dates back centuries, however, the position of a sound designer emerged approximately 50 years ago with the advancement of audio and recording technology[2]. Sound design is the art of creating and manipulating sound for various media productions, such as film, advertising, and interactive learning tools [48], which is a broad and evolving field that requires both technical skills and sensitive vision [34]. It requires significant human effort to leverage sound effects to enhance the mood, atmosphere, realism, and storytelling of different visual media [8].

Typically, sound design involves understanding the visual content, *i.e.* the semantics of objects and scenes; recording sounds from various sources, such as field recording, sampling, synthesis, or using sound libraries, for initialization of the soundtrack for the visual media; editing and manipulating sounds to make them fit the desired visual context; mixing and balancing sounds to create a coherent and immersive audio experience[3]. The cost of professional sound design is high. For example, adding sound effects to a game project requires 48 hours of sound design work, with an average sound designer rate of $30 to $100 per hour[4]; the sound design costs are expected to fall in the range from $1,440 to $4,800.

On the other hand, when an amateur aspires to create a video enriched with sound effects, he/she may encounter constraints, such as having to choose from a limited repository of sound effects offered by platforms (*e.g.*, TikTok), without the ability to generate sounds that are optimal to their specific recordings. Alternatively, they could use text-prompted sound effects generation applications (*e.g.*, Pika), which may require multiple attempts with different descriptions and subsequently editing the results, *e.g.*trimming the waveform to have an appropriate segment to match the video content. These processes can be cumbersome and limit creative freedom. Therefore, we desire to explore the possibility of devising a computational approach to sound design.

### 2.2 Vision-to-Sound

With advancements in technology and research in multimedia, human-computer interaction, computer vision, and graphics, the concept of converting visual information into sound has been implemented in recent years. Previous works on this topic can be divided into two categories: natural sound generation and music generation. As music is commonly used to enhance the atmosphere of the visual content [43] and our goal is to generate authentic sound effects for video, our discussions in this section will focus on natural sound generation endeavors.

---

[1]https://pika.art/

[2]https://www.nfi.edu/sound-design/
[3]https://www.coursera.org/courses?query=sound%20design
[4]https://www.visartech.com/blog/sound-effects-in-games-development/

For physics-based techniques, they aimed to improve the audio-visual consistency of computer-animated phenomena, *i.e.* synthesizing natural sounds for rain [28], fire [46], rigid bodies [9, 42], etc. Moreover, image sonification techniques synthesize auditory interpretations of visual stimuli, *e.g.*, mapping pixel value to various sound parameters (*e.g.*, pitch and loudness), thus enhancing scientific discovery [48] and visual accessibility to people who have visual impairments [22]. A similar idea to our *AutoSFX* is Audible Panorama [20], where suitable sound effects are chosen from a pre-collected dataset for objects in static panorama images, considering object depth and audio source placement.

On the other hand, several promising works have emerged, leveraging deep learning techniques to generate waveforms for specific objects [5, 6, 23, 29, 35, 51] and soundscapes of wild scenes [21, 50, 53]. More recently, diffusion-based models have achieved state-of-the-art sample quality in the field of audio generation. For example, Luo *et al.* [31] proposed to train a latent diffusion model with temporally and semantically aligned features on spectrogram latent space. Liu *et al.* [26], Yang *et al.* [45], and Ghosal *et al.* [16] leveraged diffusion models to generate sound conditioned on a text prompt. While these methods strive to provide precise representations of real-world soundscapes, their application is constrained to short video clips, rather than the diverse and vast videos uploaded online.

Additionally, the realm of text-to-audio generation has captured significant interest [2, 7, 25]. However, the challenge of automatic sound design differs from these works – it involves not only encoding visual content and producing a corresponding soundtrack with multiple waveforms, but also performing seamless transitions and harmonious blending.

### 2.3 Segment Anything

Segment-Anything Model (SAM) [24], a large foundation model trained on one billion masks from 11M images, supports flexible and interactive prompts in real time to achieve image segmentation. SAM represents a revolutionary method for image segmentation, demonstrating remarkable generalization capabilities when applied to different datasets. There have been recent proposals for extensive applications based on SAM [49], such as image editing [47] and style transfer [30]. Such visual segmentation techniques have inspired researchers to integrate SAM into audiovisual segmentation pipelines. For example, Mo *et al.* [32] and Wang *et al.* [44] proposed to extract visual features by utilizing the pre-trained image encoder in SAM and aggregating them with auditory features, *i.e.* following the encoder-fusion-prompt-decoder paradigm. Liu *et al.* [27] simply employed adapters to inject the audio information into the pre-trained SAM, achieving deep audio-visual fusion in the encoding stage. Following the bidirectional generation strategy, Hao *et al.* [17] proposed to use the generated segmentation masks to reconstruct audio features and minimize auditory reconstruction errors during training. Inspired by these studies, we propose to leverage SAM as the fundamental feature extraction model to contribute to audiovisual representation learning and further design a diffusion-based model to generate sound effects for visual objects.

### 3 OVERVIEW

As shown in Fig. 1, *AutoSFX* contains two main components:

A *Sound Generation* module, which formulates the visual-guides sound generation task as a collaborative parallel generation problem for both auditory and visual channels. In the training Dataset $D = \{X_i, y_i\}$, $X = \{x^v, x^a\}$ represents the video with continuous frames ($x^v$) and the corresponding audio clip's spectrogram ($x^a$), and $y_i$ represents the ground truth visual segmentation masks. Given the input data $X$, the goal of the *Vision-to-Sound* module is to generate the audio spectrogram with the guidance from $x^v$. Moreover, $i$ represents the sample index for a clearer explanation.

A *Sound Optimization* module, designed with constraints for seamless transition and harmonious mixing, aims to synchronize the generated sounds with video frames to create an immersive audiovisual experience. Let $A_k = \{a_1, a_2, ..., a_k\}$ denote the generated sounds for the sounding object $k$ in the video clip. We choose the optimal $a$ that minimizes the cost:

$$C_{\text{total}}(A, x^v) = C_{\text{tran}}(A_k, A_{k+1}) + \lambda C_{\text{mix}}(A_k, x^v), \quad (1)$$

where $C_{\text{tran}}(\cdot)$ is the transition cost term, which measures the transition difficulty between two continuous sounds, constraining the pitch and tempo. $C_{\text{mix}}(\cdot)$ is the cost term for evaluating consistency between the visual information and the generated results based on the estimation of depth and emotion expression. $\lambda$ is a regularization factor to balance these two terms.

### 4 SOUND GENERATION

As shown in Fig. 2, the *Sound Generation* module has three parts: First, we encode the visual and auditory representations from two branches and fuse them in an attention manner; Second, we resort to the diffusion model to map the audiovisual information to latent representation and simultaneously to help model the conditional distribution; Finally, spectrograms could be generated with the projection of the visual prompt into the auditory space.

### 4.1 Audiovisual Fusion

For the visual branch, we sampled the input videos at intervals of 1 second to obtain frames $x^v \in \mathbb{R}^{T_v \times 3 \times H \times W}$, where $T_v$ represents the number of frames. We extract visual features from the ImageNet pre-trained SAM backbones, which is based on ViT [12] and consists of 12 transformer layers. We also apply the visual encoder with bottleneck adapters [19] and obtain the visual feature $F_v \in \mathbb{R}^{d_v \times H \times W}$.

On the other hand, as a spectrogram (*i.e.* 1-channel 2D images) autoencoder with reconstruction objective as self-supervision has demonstrated the effectiveness of image-to-audio generation [44], we transfer the audio mono-waveforms (with a sampling rate of 16kHz) into a sequence of mel-spectrogram sample $x^a \in [0, 1]^{C_a \times T_a}$, where $C_a$ denotes the mel channels and $T_a$ is the number of frames. Then, we leverage the VGGish [18], a model designed for capturing both temporal and spectral information, to extract auditory features $F_a \in \mathbb{R}^{T_a \times d_a}$. $d_a$ is 128 as the default.

$F_a$ includes auditory guidance for sound objects that is crucial for image segmentation, and $F_v$ contains important visual context information for consistent audiovisual sound generation. However, $F_a$ and $F_v$ extracted from different branches do not align well. To address this, we apply an audiovisual cross-modal attention module, shown in Fig. 3, to align the audio sources and visual locations

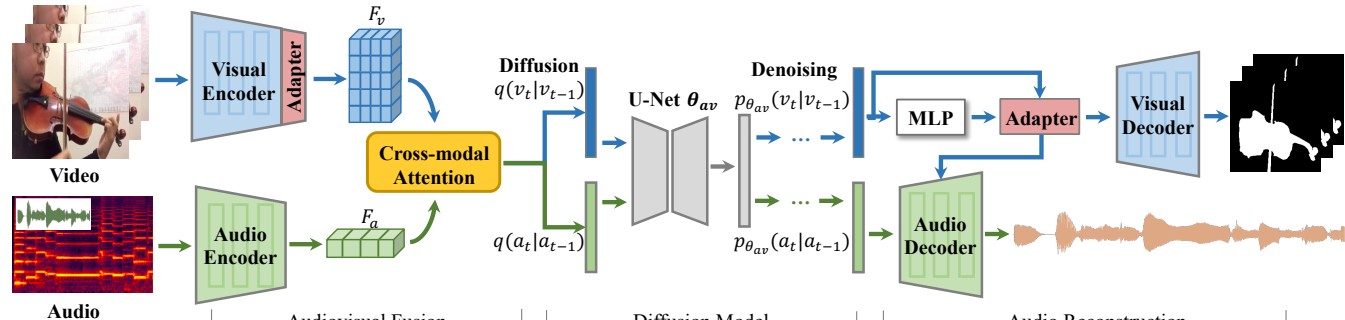

Figure 2: Model architecture of the *Sound Generation* module in *AutoSFX*.

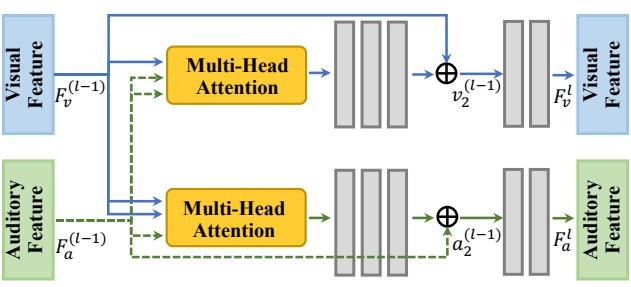

Figure 3: *AutoSFX*'s attention architectures for audiovisual alignment.

by treating them as a joint attention space. The attention module takes $F_v$ and $F_a$ as inputs. In the joint attention setting, attention operates simultaneously over time and space. Moreover, we utilize the multi-head attention (MHA) layer [41] to capture different aspects of the input, allowing for a more comprehensive and nuanced representation of the relationships between $F_v$ and $F_a$. At the $l$-th layer, when considering a auditory query, the directional attention operations can be described as:

$$a_1^{(l)} = \text{MHA}(F_v^{(l-1)}, F_a^{(l-1)}),$$
$$a_2^{(l)} = \text{LN}(a_1^{(l)} + F_a^{(l-1)}),$$
$$F_a^{(l)} = \text{LN}(f(\text{Dropout}(a_2^{(l)})) + F_a^{(l-1)})).$$

For the auditory direction, we modulate the visual features $v^{(l)}$ using the auditory features $a^{(l)}$ by swapping $F_v$ and $F_a$ in the above equation.

## 4.2 Diffusion-Based Vision-to-Sound

Diffusion-based models, first transform a given data distribution into unstructured noise (usually Gaussian noise) and then proceed to learn how to reverse the forward process to recover the original data distribution. In this paper, we deploy an AV-Diffusion Model to bidirectionally generate we further propose a bidirectional multimodal latent diffusion model, shown in the middle part of Fig. 2. Our goal is to recover two consistent modalities within one diffusion process. Specifically, the forward diffusion maps both audio and video data to noise independently, while the reverse process gradually reconstructs the original multimodal contents using a

unified model. With the paired visual and auditory feature $(a, v)$ (*i.e.* the simplification of $F_a^l$ and $F_v^l$), the forward processes of each modality are independent. Taking the auditory feature $x_a$ as an example, the corresponding forward process is defined as:

$$q(a_t | a_{t-1}) = \mathcal{N}_a(a_t; \sqrt{1 - \beta_t} a_{t-1}, \beta_t \mathbf{I}).$$

$t \in [1, T]$ is the time step. The forward process for visual represents $v$ a similar formulation. Let $a_t$ as the sample fitting standard Gaussian distribution and is independent from $a_0$ using the Markovian forward process. We can calculate any $a_t$ through:

$$q(a_{1:T} | a_0) = \sum_{t=1}^{T} q(a_t | a_{t-1}).$$

On the other hand, we utilize the joint reconstruction of audiovisual pairs from independent Gaussian distributions [36]. The coupled U-Net [36] $\theta_{av}$ takes both auditory and visual information as inputs and reinforces generation quality for each other. Specifically, the reverse process $p_{\theta_{av}}(v_{t-1} | (v_t, a_t))$ for obtaining $v_{t-1}$ in visual domain is defined as:

$$p_{\theta_{av}}(a_{t-1} | (v_t, a_t)) = \mathcal{N}(a_{t-1}; \mu_{\theta_{av}}(v_t, a_t, t)),$$

where $a_{t-1}$ is generated from a Gaussian distribution jointly determined by both $v_t$ and $a_t$. The reverse process for visual represents $v$ a similar formulation.

**Visual-Promted Sound Generation.** As declared in previous works [35], decoding in the auditory space with visual prompts can enhance the model's generalization ability. Our visual-promoting module was designed to prompt the audio decoder to generate sounds that are consistent with the objects/events from the visual space. As shown in the third part of Fig. 2, we first feed $v$, *i.e.* the output of the visual branch from the coupled U-Nets, into an MLP. We then leverage a bottleneck adapter $\text{ColA}(\cdot)$ to model the audiovisual correlation, yielding the updated visual feature $F_c \in \mathbb{R}_v^d$:

$$F_c = \text{ColA}(\text{MLP}(v)) + \text{MLP}(v).$$

As the visual prompt, $F_c$ is further fed into the audio decoder. We adopt WaveNet [40] as the decoder to convert the synthesized spectrogram into waveform $A$. Simultaneously, we also obtain the final visual output, *i.e.* mask embedding $F_m \in \mathbb{R}^{d_v \times H \times W}$, which is then upscaled by convolutional blocks.

## 4.3 Training

**Learning Objectives.** *AutoSFX* highlights the potential opportunities arising from the visual segmentation into the audio generation pipelines. The total loss has three terms:

i) *Reconstruction Loss*, i.e. the L2 error for measuring the generated spectrogram $s'$ and the ground truth $s$:

$$\mathcal{L}_{recon} = ||s' - s||_2^2;$$

ii) *Segementaion Loss*, we use the binary cross-entropy $BCE(\cdot)$ loss to measure the difference between the predicted mask $M_p$ and the ground truth $M_g$:

$$\mathcal{L}_{seg} = \text{BCE}(M_p, M_g);$$

iii) *Generation Loss*, is defined as the mean squared error in the noise space $\epsilon \sim \mathcal{N}(\mathbf{0}, \mathbf{I})$. We leverage $\epsilon$-prediction to optimize the network, i.e.

$$\mathcal{L}_{\theta_{av}} = \mathbb{E}_{\epsilon \sim \mathcal{N}_{\{0, I\}}} \left[ \lambda_t ||\tilde{\epsilon}_\theta(a_t, v_t, t) - \epsilon||_2^2 \right], t \in [0, T],$$

where $t$ represents a random term for stochastic gradient descent and $\lambda_t$ is an optional weighting function.

Therefore, the final loss function is

$$\mathcal{L}oss = \lambda_1 \mathcal{L}_{recon} + \lambda_2 \mathcal{L}_{seg} + \lambda_3 \mathcal{L}_{\theta_{av}}. \tag{2}$$

During the training phase, we only fix the parameters of both visual and auditory encoders, and update others, e.g., the adapters, mask decoder, and spectrogram decoder.

**Dataset.** We leverage AVSBench [52], a recently released video segmentation dataset, providing masks for sounding objects with audio signals, to train our *Sound Generation* module. It covers 23 object categories, e.g., animal and human-related sound events. Specifically, the single-source subset, includes 4932 videos with 10,852 annotated frames.

## 5 SOUND OPTIMIZATION

When a soundtrack is out of sync with the visual content, such as lagging behind or rushing ahead of the visual elements, it will result in a discordant, disjointed, or confusing viewing experience [38]. As a result, sounds should be both temporally and content-wise aligned with visual content, i.e. maintaining consistent changes when visual content changes and sounds played at the same time should be mixed properly.

To improve the consistency between the generated audio and input video, we formulate the synchronization process as an optimization with several constraints. As depicted in equation 1, we design two cost terms, i.e. transition cost and mixing cost.

### 5.1 Transition Cost

A key observation of our work is that a transition from sound effect $a_k$ to another one $a_{k+1}$ sounds natural when pitch and tempo are similar [43]; and if the pitch is harmonious progressed. We compute the transition cost $C_{tran}(\cdot)$ by combining these two cost terms:

$$C_{tran}(A_k, A_{k+1}) = \alpha_1 C_{tem}(A_k, A_{k+1}) + \alpha_2 C_{pit}(A_k, A_{k+1}),$$

where $C_{tem}(\cdot)$ is *tempo cost*, evaluating the BPM (beats per minute) difference between two sounds. For *pitch cost*, $C_{pit}(\cdot)$, we estimate pitch differences by computing chroma features [13] for each beat

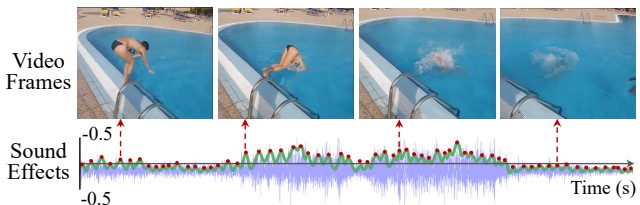

**Figure 4: Example video frames and the corresponding sound effect's peak.**

in a sound clip, respectively. Then we compute the average cosine distances between $A_k$ and $A_{k+1}$.

### 5.2 Mixing Cost

Driving content-wise audiovisual consistency, is not merely aligning sounds with visual elements; it is more about creating an immersive experience that resonates with the audience on a deeper level [6]. So we delve into mixing techniques commonly used in the sound design process, meticulously analyzing and integrating them to form the mixing cost $C_{mix}(A_k, x^v)$. Below we describe how mixing guidelines are formulated and adapted to our setting.

**Rhythm.** Video with a greater rhythm-matched soundscape is preferred in the viewing experience. We design a cost term, $C_r(A_k, x^v)$, to find the best part of $A_k$ that matches the visual content progression. This cost has two elements:

i) *Speed/Tempo*, denoted by $C_s(A_k, x^v)$, influences the sense of urgency, excitement, or contemplation, etc.: the faster the speed, the higher the content tension. We apply the visual beats extraction approach [11] to obtain the visual tempo of $x_v$, and calculate the absolute distance between it and the $A_k$'s BPM as $C_s(\cdot)$.

ii) *Movement*, whether it's the action of subjects or the camera's motion, the consistent sound effects can significantly influence the rhythm of the video. We believe that changes in motion should consistently match the peak changes in sound effects. For each segmented object, we first compute the homography transformation matrix linking consecutive frames $f$ and $f'$. We then use the estimated homography matrics $H(f, f')$ and $H(f', f'')$ from three consecutive frames to compute the movement $mov(f)$ for frame $f$:

$$mov_v(f) = \frac{1}{4} \sum_{n=1}^{4} ||H(f', f'')p_{f'}(n) - p_{f'}(n) - (H(f, f')p_f(n) - p_f(n))||_2,$$

where $p_f(i), i \in [1, 4]$ represents the 2D positions of the four corners of the sounding objects measured in pixels. For the sound effect, we find $A_k$'s peak (a long-pitch plateau forming a flat turn) $mov_s(f)$ by binary search. Consequently, the movement cost term $C_m(A_k, x^v)$ is defined as the Kullback–Leibler divergence between $mov(f)$ and $mov(p)$, i.e.

$$C_m(\cdot) = \sum_{f \in F} mov_v(f) ln \frac{mov_v(f)}{mov_s(f)}. \tag{3}$$

As shown in Fig. 4, we demonstrate the comparison between video keyframes and the corresponding sound effect's peak.

**Emotion.** To address the significant gap between the visual domain and the audio space, we leverage the color tone and frequency distribution representing visual and auditory information, respectively. These attributes have been demonstrated to be tightly

related to emotion expression in video creation [3, 39]. For example, low-frequency sounds might be more likely to induce sadness, while high-frequency sounds can invoke happiness or excitement. We first use color histograms to obtain the saturation and brightness of frames and further compute the ratio $\xi_v(f)$ of pixels within high saturation and brightness regions, *i.e.* above 70% of the maximum value (*i.e.* 255). Similarly, we compute the ratio $\xi_a(f)$ of high-frequency regions, *i.e.* above 30% of the maximum value of $A_k$. Therefore, $C_e(A_k, x^v)$ is defined as:

$$C_e(\cdot) = ||\xi(f) - \xi(f)||_2,$$

**Distance.** When mixing multiple sound effects that playback overlapped, the relative depth should be considered – resulting in a layered soundscape that enhances the perception of distance and space within the audiovisual content [20]. We apply the approach proposed by Dai *et al.* [10] to estimate the segmented objects' three translations in real time and calculate Euclidean distance between each pair of sounding objects, yielding $d_v(p, p + 1), p \in [1, N]$. For sound effects, we compute the corresponding loudness distance, *i.e.* $d_a(A_p, A_{p+1})$. We normalize $d_v$ and $d_a$ into $[-1, 1]$, and define the distance cost as:

$$C_d(\cdot) = -\frac{d_v(p, p + 1)}{d_a(A_p, A_{p+1})},$$

The goal of harmonious mixing is to find a sequence of generated sound effects $A_k$ by selecting and ordering a subset of candidate sounds, cutting within them, performing low/high-frequency pass, and adjusting the loudness. The total mixing cost is

$$C_{mix}(A_k, x^v) = \beta_1 C_r(A_k, x^v) + \beta_2 C_e(A_k, x^v) + \beta_3 C_d(A_k, x^v),$$
$$C_r(\cdot) = C_s(A_k, x^v) + \gamma C_m(A_k, x^v).$$

$\beta$ and $\gamma$ are regularization factors to balance these cost terms.

### 5.3 Optimization

Since our optimization problem is combinatorial and the number of combination items can vary, we adopt the Reversible Jump MCMC (RJMCMC) framework to explore the space of possible soundtrack extensively. To efficiently explore the solution space, we apply the simulated annealing process in the optimization process. We define a Boltzmann-like objective function:

$$f(A^*) = \exp(-\frac{1}{t}C_{total}(A, x^v)), \qquad (4)$$

where $t$ is the temperature of the simulated annealing process, which decreases gradually throughout the optimization. There are three types of moves that can be selected by the optimizer:

(1) *Modify Sound*, including modifying the tempo, pitch, and loudness of the current sound effect. Take tempo as an example, the modification is defined as $tem' = tem_0 + \triangle tem_0$, where $\triangle tem_0$ is sampled from a Gaussian distribution whose mean is zero and variance is $0.1 tem_0$;

(2) *Change Onset*, *i.e.* randomly change the timestamp of playing the current sound effect;

(3) *Swap Sound*, *i.e.* randomly change to another sound clip from the generated sounds.

The selection probabilities of the moves are $p_m$, $p_c$, and $p_s$, which are set as 0.35, 0.3, and 0.35. Compared to existing sound effect generation approaches discussed in § 2.2, this hybrid optimization approach not only ensures the diversity of sound styles, but also preserves seamless transition between sounds and harmonious mixing consistent with the visual content.

## 6 AUTOSFX INTERFACE

As shown in Fig. 1, we develop an interface for video creators or sound designers to facilitate automatic sound generation for videos. Specifically, The *Operation Panel* offers users three options: "Chapters", displaying keyframes of video clips and allowing for quick skipping; "Objects", showing the segmentation heatmap of each video clip and enabling users to specify their desired sounding objects; "Sound Effects", presenting our generated and optimized results that match the visual content. Please refer to the demo for detailed operation of our *AutoSFX*.

## 7 EXPERIMENTS

In this section, we first introduce the experiment setup (§ 7.1), including datasets, evaluation metrics, and implementation details. Then, we evaluate the performance of our *AutoSFX* in comparison to state-of-the-art methods for vision-to-sound (§ 7.2). We also provide mass generated results and conduct user studies to evaluate our generated sounds and the interface (§ 7.3).

### 7.1 Setup

**Dataset.** We leveraged two datasets to evaluate the performance of the sound generation of our proposed *AutoSFX*.

i) VEGAS [53], containing 28,109 videos with both visual and auditory information, is derived from AudioSet [15]. It covers 10 categories, such as dog barking, baby crying, and water flowing.

ii) VGGSound [4], contains over 200k clips for 309 different sound categories, which are all downloaded from YouTube. Each clip lasts at least 10 seconds and the corresponding label is flat (*i.e.* no hierarchy architecture among labels).

**Evaluation Metrics.** We quantified the performance by adopting audio generation metrics outlined in [6, 29].

i) Quantitative evaluation involves: i) *missing sound* (Miss. S.), *i.e.* when an event should produce sound but the model fails to generate it; ii) *redundant sound* (Red. S.), the opposite of missing sound, *i.e.* the model generates sound without any corresponding visual stimuli; iii) *mismatched sound* (Mism. S.), *i.e.* the generated sound does not match the video content; iv) *objective similarity grade (OSG)*, *i.e.* measuring the distance between ground truth and the generated audio at the acoustic feature level.

ii) User study, primarily focuses on participants' audiovisual experience in user-uploaded YouTube videos with soundtracks designed by different approaches, including our *AutoSFX*, online tools, and professional sound design.

We also conducted comparisons between our *AutoSFX* and audiovisual segmentation approaches, demonstrating a comparable capability on the task of video segmentation. Please refer to supplementary materials for more details.

**Implementation Details.** We resized all video frames to a size of $1024 \times 1024$ and extracted the log Mel-Spectrogram using 64 mel filter banks over 1 second of audio data sampled at 16,000 kHz on the training set of VEGAS and VGGSound. The weight for balancing the three losses in Equation 2 was empirically set to 1. We employed

**Table 1: Quantitative results of models trained on two different settings. The lower the value, the better the performance.**

| VEGAS | Miss. S. | Red. S. | Mism. S. | OSG |
|---|---|---|---|---|
| Chen *et al.* [6] | 15.67% | 16.26% | 17.39% | 11.2947 |
| Liu *et al.* [29] | 9.22% | 10.30% | 12.06% | 9.8235 |
| ***AutoSFX* (Ours)** | **8.08%** | **10.27%** | **10.18%** | **9.0012** |
| VEGAS & VGGSound | Miss. S. | Red. S. | Mism. S. | OSG |
| Iashin *et al.* [21] | 11.37% | 19.38% | 24.22% | 17.2234 |
| Lou *et al.* [31] | 8.81% | 9.22% | **14.02%** | **12.0321** |
| ***AutoSFX* (Ours)** | **7.44%** | **7.93%** | 15.11% | 13.3287 |

Adam optimizer to optimize the model parameters with an initial learning rate of $10^{-4}$ with cosine decay. The batch size was defined as 8 and we trained for 40 epochs. Additionally, we utilized PyTorch for model training on an NVIDIA A800 GPU.

## 7.2 Quantitative Evaluation

To fairly compared our *AutoSFX* and other vision-to-sound models, we trained our diffusion-based *Sound Generation* module under two settings: i) When comparing to models [6, 29], we utilized 80% of the data from the VEGAS dataset to perform the training process. The remaining 20% data was used for evaluation. Each video clip has three corresponding sound effects generated by different models; ii) When comparing to models [21, 31], we trained all models on both VEGAS and VGGSound datasets. The same splitting strategy was used for training and evaluation. Note that, some of the compared models are not official implementations; however, we have made efforts to reproduce their pipeline.

For the metrics of *missing sound*, *redundant sound*, and *mismatched sound*, we recruited 10 participants to evaluate the generated results of different models. The objective similarity grade (OSG), consisting of maximum modulation spectrum (MMS), mean spectral flux (MSF), and root mean square (RMS), is defined as:

$$OSG = r_1 \cdot MMS + r_2 \cdot MSF + r_3 \cdot RMS, \tag{5}$$

where $r_1 = 0.57, r_2 = 0.52, r_3 = 0.45$, as suggested in [29].

**Results and Analysis.** As shown in Table 1, we demonstrate the evaluation results of different models under the two settings. The lowest the value, the higher the performance. Overall, our *AutoSFX* attains the lowest error for all metrics under the first setting. Similar to the statistical results demonstrated in [6], the sound generation of "sneeze" and "Cough" demonstrated relatively lower performance, leading to the suboptimal results in our experiments.

For the results of the second setting, the model proposed by Iashin *et al.* [21], which generates sound effects with a fixed length of 10 seconds, yielded significantly lower performance than that of the other two approaches, particularly for the *mismatched sound*. On the other hand, our results are comparable to those of Diff-Foley (*i.e.* Lou *et al.* [31]) – for example, *AutoSFX* outperformed in animal sounds, while Diff-Foley excelled in instrument sounds. We believe that such results could be improved with our *Sound Optimization* module. In Fig. 5, we illustrate some sound effects generated by different approaches along with the alignment cases, *i.e. missing*

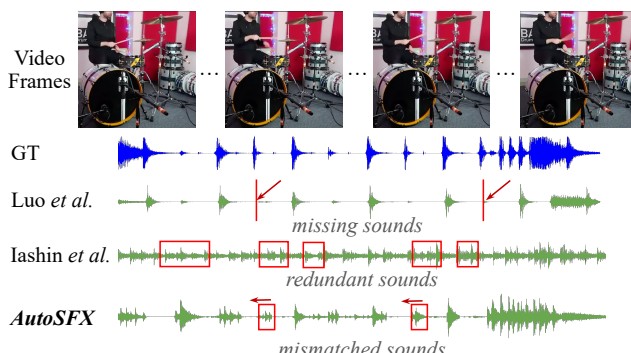

**Figure 5: Examples of *missing sound*, *redundant sound*, and *mismatched sound* in generated sound effects.**

sound, *redundant sound*, and *mismatched sound*. We also provided some sounds generated by different approaches in our demo.

## 7.3 User Study

In addition to the quantitative evaluation, we further evaluated the effectiveness of our *AutoSFX* for practical sound design scenarios.

**Different Approaches.** We compared three approaches:

- *AutoSFX*, our generated and optimized sound effects;
- Pika[5], an online tool for users to generate sound effects based on generated videos or text prompts;
- professionally designed soundtrack by sound designers.

**Dataset.** We randomly selected 10 videos on YouTube from the "Sound Design Tutorial" category. These videos cover different objects, styles, and themes, typically uploaded by professional sound designers. Therefore, we leveraged the provided soundtrack as the professional results. We then applied *AutoSFX* and Pika to automatically generate sound effects for these videos. Please refer to the demo for videos featuring different soundtracks "designed" by different approaches.

**Participants.** We recruited 30 participants aged between 30 and 32, who reported normal or corrected-to-normal vision, with no color blindness and normal hearing. 10 of them are video creators who usually record, edit, and upload videos at least once a week (referred to as *Creators*); and the rest are viewers of video streaming platforms, like YouTube and TikTok (referred to as *Viewers*).

**Procedures.** The goal of this study is to evaluate how well the soundtrack matches the visual content. We asked participants to rate the authenticity of the sounds, audiovisual temporal consistency, sound quality, and overall viewing experience, using a 1-5 Likert scale, where 1 indicates poor performance and 5 indicates the opposite. For each video, the three corresponding soundtracks were presented randomly so as to avoid bias. Participants were allowed to view each video and play the corresponding soundtrack an unlimited number of times. Finally, all participants were invited to use our implemented interface. We then conducted a semi-structured interview about their experience to explore nuanced insights into our system design.

---

[5]https://pika.art/

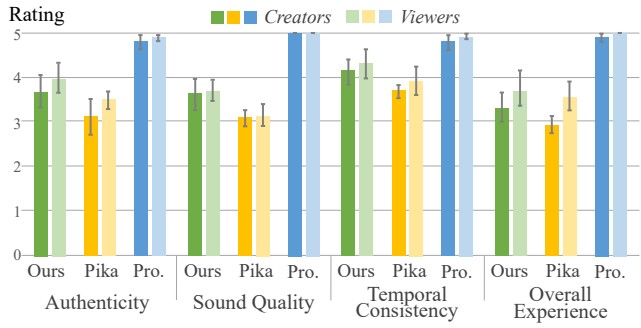

**Figure 6: Participants' ratings of sound effects generated by different approaches, *i.e.* our *AutoSFX*, an online tool (Pika), and professional sound design. We grouped ratings by *Creators* and *Viewers*.**

**Results and Analysis.** As shown in Fig. 6, we demonstrate the ratings by different participant groups, *i.e. Creators* and *Viewers*.

*Authenticity and Sound Quality*, *i.e.* whether the sounds authentically and plausibly express the video content. Overall, ratings for all three approaches average around 4, with the professional results exceeding 4.9. Our *AutoSFX* received an average rating of $M = 3.92, SD = 0.76$ over all participants, *i.e.* $M = 3.68, SD = 0.86$ for *Creators* and $M = 3.95, SD = 0.79$ for *Viewers*. According to participants' feedback, over 75% of our generated sound effects for visual objects were largely successful in tricking humans into thinking that these sounds were real. However, the ratings for Pika's results are relatively lower, yielding an average of $M = 3.41, SD = 0.92$. Such results could be explained by the fact that sound effects used by professional designers are recorded or created using foley techniques; whereas the other two kinds of sounds are generated. On the other hand, when multiple objects appear in the video, Pika's results sometimes sound noisy, not to mention a soundscape with depth distinctions; while our *AutoSFX* could generate soundS for each segmented object and mix them through the optimization.

Similar ratings were received for the ratings of sound quality. When we asked participants about the "fake" sounds or low-quality ones, we found there were two main reasons for their judgments:

- Sounds can potentially convey significant information. Take the baby crying as an example; it often indicates that the baby has a need or discomfort that must be addressed. However, the generated sounds may not show diverse variations.
- Sound quality significantly influences the perception of authenticity – whether for music, animal sounds, or even human speeches.

*Audiovisual Temporal Consistency.* The professionally designed sounds received the highest ratings, *i.e.* an average of $M = 4.89, SD = 0.10$, followed by the results by our system ($M = 4.16, SD = 0.97$). On the contrary, Pika's results showed relatively lower performance ($M = 3.94, SD = 1.05$). Note that, the length of sound generated by Pika is longer than that of the video, so we trimmed sounds to start and stop with the video. Additionally, the ratings of *Creators* are significantly lower than those of *Viewers*. Some *Creators* reported being more rigorous on temporal consistency. We also observed that these participants usually repeatedly played video segments to check the alignment.

*Overall Experience.* To verify the above three factors contributed to the overall experience, we computed Bivariate (Pearson) correlation coefficients between their ratings and overall ratings, respectively. As a result, we obtained positive correlations, *i.e.* $r = .51, p < .05$ (authenticity), $r = .44, p < .05$ (temporal consistency), and $r = .39, p < .05$ (sound quality). This further supports the effectiveness of our adopted *Sound Optimization* module. According to the participants' feedback, we observe that while some rated authenticity, synchronization, and quality highly, they gave lower ratings for overall consistency. They explained that sometimes the high-frequency part of the generated sound is unclear, leading to poor viewing experience.

**About the Interface.** After participants used our interface to generate sound effects for videos, we asked them about their experience by giving a rating of "Not Helpful", "Somewhat Helpful", and "Very Helpful". 80% (16 out of 20) of the *Viewers* rated it as "Very Helpful", noting its significant helpfulness for amateur creators to quickly produce videos with an appealing soundscape. 30% of *Creators* considered the system "Very Helpful" as they appreciated the tool's ability to automate the complex process of sound selection, editing, and mixing; 60% found it "Somewhat Helpful," reducing some efforts to produce videos; only two participants rated is as "Not Helpful" due to the sound quality issues. All participants expressed a desire for a more comprehensive system like the Adobe series to make such automation widely applicable.

## 8 CONCLUSION

Conditioned on visual content, automatic sound design is challenging but can streamline the video-making process, benefiting video creators and sound designers. In this paper, we propose a computational approach to automatically generate sound effects for videos considering audiovisual consistency. Our *AutoSFX* is the first attempt to integrate sound generation techniques into practical sound design scenarios.

*Limitation and Future Work.* Videos, recorded by different creators may have different intended uses, themes, and styles featuring different backgrounds, contextual storytelling, lighting, etc. Due to the difficulty of learning effective representations of these factors by computer vision techniques and obtaining accurate annotations, we only generate sound effects for audiovisual consistent objects and involve quantifiable cost terms during optimization. Moreover, when dealing with intricate scenarios, *e.g.*, a crowded concert with numerous individuals and instruments, our *Sound Generation* module may encounter challenges in processing such visual information. We also provide some failure examples of such issues in supplementary materials.

Currently, our *AutoSFX* primarily focuses on generating sounds for realistically sounding objects. However, sound design often utilizes abstract sounds to enhance the atmosphere, emphasize emotions, and convey subtle nuances. For example, a "woosh" sound could be used to describe the camera movement. Inspired by text-prompt techniques, *e.g.*, text-guided music generation [1, 33], future work could further incorporate text-prompt during the training process of the generation module. We also believe there is room for improvement in video understanding and in particular, generating abstract, indescribable, and context-ware sound effects.

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

Received 20 February 2007; revised 12 March 2009; accepted 5 June 2009

