# OpenReview forum: "AutoSFX: Automatic Sound Effect Generation for Videos"
_acmmm.org/ACMMM/2024/Conference — MM2024 Poster_

### Official Review · Reviewer_yrQ1 · 2024-05-24

**Rating:** 4
**Confidence:** 3

**Summary:**

The proposed system, AutoSFX, aims to significantly streamline and broaden the applicability of automating sound design for videos. The key insight behind AutoSFX is that ensuring consistency between auditory and visual information, performing seamless transitions between sound clips, and harmoniously mixing sounds playing simultaneously is crucial for creating a unified audiovisual experience. AutoSFX capitalizes on this concept by aggregating multimodal representations through cross-attention and leveraging a diffusion model to generate sound with embedded visual information. Additionally, AutoSFX optimizes the generated sounds to render the entire soundtrack for the input video, leading to a more immersive and engaging multimedia experience.

**Strengths:**

AutoSFX introduces a new method for automated sound design in videos. It combines visual and auditory information using cross-attention and a diffusion model. This innovative approach leverages the Segment Anything Model (SAM) for audiovisual fusion, marking a significant breakthrough in the field. AutoSFX is thoroughly evaluated using both objective and subjective methods. The system's performance is validated on widely recognized datasets such as VEGAS and VGGSound, demonstrating its capability to generate high-quality sound effects. The paper clearly articulates the motivation, methodology, and contributions of AutoSFX. Detailed descriptions of the system architecture, including the Sound Generation and Sound Optimization modules, provide clarity on how the system functions.

**Limitations:**

The paper introduces innovative sound generation techniques for practical sound design. However, it heavily relies on existing models and techniques, such as the Segment Anything Model (SAM) and diffusion models, which raises concerns about the originality of the methodology.

**Suitability:**

3

---

### Official Review · Reviewer_164r · 2024-05-24

**Rating:** 4
**Confidence:** 4

**Summary:**

This paper proposed a sound effect generation system, AutoSFX, to automatically generate sounds guided by visual information. The proposed framework integrates multimodal representations via cross-attention and leverages a diffusion model to generate sound conditioned on visual cues. A user-friendly interface is also developed for the application. Experiments on the VEGAS and VGGSound test set verify the effectiveness of the proposed method.

**Strengths:**

1. Considering segmentation characteristics, such as leveraging SAM as the visual backbone, introducing segmentation loss, and training on the AVSBench dataset is interesting in the music/sound generation research community.

2. The authors conduct comprehensive qualitative and quantitative results to show the proposed algorithm outperforms previous methods. The user-friendly interface also makes the system applicable.

3. The writing is clear and easy to follow.

**Limitations:**

1. Technical contribution is limited. Leveraging a latent diffusion model for generation tasks is not novel, and the audiovisual fusion/cross-attention module is also utilized in many previous audio-visual methods.

2. For the visual encoder, the authors select the recent advance, SAM, yet VGGish is adopted as the audio encoder. Why use VGGish rather than some more recent works such as Beats or AST? Is there any ablation? Besides, for the visual encoder, is there any ablation to show its effectiveness?

3. The authors analyze the audiovisual consistency, yet some existing music-video corresponding metrics need to be compared and discussed, such as beat hit rate, and beat coverage rate proposed in [1, 2]. For the visual rhythms, some previous works apart from [4] are also in need of discussion [2, 3].


Reference:

[1] Zhu, Ye, et al. "Discrete contrastive diffusion for cross-modal music and image generation." arXiv preprint arXiv:2206.07771 (2022).
[2] Yu, Jiashuo, et al. "Long-term rhythmic video soundtracker." International Conference on Machine Learning. PMLR, 2023.
[3] Su, Kun, Xiulong Liu, and Eli Shlizerman. "How does it sound?." Advances in Neural Information Processing Systems 34 (2021): 29258-29273.
[4] Davis, Abe, and Maneesh Agrawala. "Visual rhythm and beat." ACM Transactions on Graphics (TOG) 37.4 (2018): 1-11.

**Suitability:**

3

---

### Official Review · Reviewer_SgaQ · 2024-05-26

**Rating:** 4
**Confidence:** 3

**Summary:**

This paper presents AutoSFX, a novel system designed to automatically generate sound effects (SFX) for videos, ensuring audiovisual consistency, seamless transitions, and harmonious mixing. AutoSFX utilizes deep learning techniques, including the Segment Anything Model (SAM) for image segmentation and a diffusion model for sound generation. The system is composed of two main modules: Sound Generation and Sound Optimization. The Sound Generation module integrates audiovisual features to produce sound spectrograms, while the Sound Optimization module ensures the generated sounds align well with the visual content. The paper reports promising results from experiments conducted on the VEGAS and VGGSound datasets and a user study evaluating the usability and effectiveness of the system.

**Strengths:**

1. Novelty: The integration of SAM for audiovisual feature extraction and the use of a diffusion model for sound generation represent innovative approaches in the field of automatic sound effect generation.
2. Technical Approach: The paper provides a detailed description of the system architecture, including the use of cross-modal attention and diffusion-based models, which are technically sound and state-of-the-art.
3. Evaluation: The system's performance is evaluated through extensive experiments on well-known datasets (VEGAS and VGGSound) and a user study, providing both quantitative and qualitative insights into its effectiveness.
4. Applications: The developed user interface and the potential applications in video creation and VR games demonstrate the practical relevance and impact of AutoSFX.

**Limitations:**

1. Complex Scenarios: The system may struggle with complex scenarios involving multiple overlapping sound sources, such as crowded events or concerts. This limitation is acknowledged but not fully addressed in the paper.
2. Abstract Sound Generation: While the system excels at generating realistic sounds, it does not handle abstract sound effects used to enhance the atmosphere or convey emotions, which are often used in professional sound design.
3. Sound Quality: Some generated sounds, particularly high-frequency components, may lack clarity, affecting the overall viewing experience. This issue was highlighted in the user study feedback.
4. Limited Contextual Understanding: The system primarily focuses on audiovisual consistent objects and might not fully capture the broader context or thematic elements of the video, which could be crucial for a holistic sound design.

**Suitability:**

3

---

### Meta-Review · Area_Chair_eJ2p · 2024-07-05

**Recommendation:** Accept (Poster)
**Confidence:** 4

**Metareview:**

The paper is written in a clear manner and well structured. All 3 reviewers recommend acceptance, 2 weak and 1 borderline accept. The authors have provided a rebuttal that improved the overall rating from 4.0 to 4.67.
One reviewer (R3) remains somewhat more critical after rebuttal. Here, there are some contradictions in the review, in "strengths" mentioning the approach to be a breakthrough, and in "limitations" stating that "AutoSFX heavily relies on existing models and techniques, such as the Segment Anything Model (SAM) and diffusion models, which raises concerns about the originality of the methodology."
All reviewers appreciate the novel approach that combines SAM and diffusion modelling in the context of audiovisual fusion. The technical approach and depth of presentation were appreciated by all, too. A further strong point mentioned by all is the systematic and extensive evaluation including also a user test.

The limitations mentioned by the reviewers were mostly well addressed in rebuttal.
Some that cannot directly be solved but were also openly reported by the authors relate to the sound quality, which may be limited in certain cases and may affect overall viewing experience (R1). Also, limitations for abstract sounds were mentioned, and for complex scenes, which cannot fully be overcome. Some improvement can be achieved with the optimization component.

Overall, I recommend accepting this paper given the ratings and especially universal appreciation of the merits, as well as a good rebuttal.
If space permits, I recommend "oral" presentation, but a "poster" could also be good.

In case of final acceptance of the paper, authors are strongly encouraged to make sure that all points mentioned in the rebuttal and further reviewer comments are considered for the camera-ready version of the paper.

- modification 08 July 2024 - updated to "accept as poster" after interaction with SAC -